# Ideas and perspectives: enhancing the impact of the FLUXNET network of eddy covariance sites

Dario Papale[1,2]

[1]DIBAF, University of Tuscia, Viterbo, 01100, Italy
5  [2]Euro-Mediterranean Center on Climate Change (CMCC), Lecce, 73100, Italy

*Correspondence to*: Dario Papale (darpap@unitus.it)

**Abstract.** In the last 20 years, the FLUXNET network provided unique measurements of $CO_2$, energy and other greenhouse gases exchange between ecosystems and atmosphere measured with the eddy covariance technique. These data have been widely used in different and heterogeneous applications and FLUXNET became a reference source of information not only for ecological studies but also in modeling and remote sensing applications. The data are in general collected, processed and shared by regional networks or by single sites and for this reason it is difficult for users interested to analysis involving multiple sites to easily access a coherent and standardized dataset. For this reason, periodic FLUXNET collections have been released in the last 15 years, every 5 to 10 years, with data standardized and shared under the same data use policy. However, the new tools available for data analysis and the need to constantly monitor the relations between ecosystems behaviour and climate change, require a reorganization of FLUXNET in order to increase the data interoperability, reduce the delay in the data sharing and facilitate the data use. All this keeping in mind the large effort made by the site teams to collect these unique data and respecting the different regional and national networks organization and data policies. Here a proposal for a new organization of FLUXNET is presented with the aim to stimulate a discussion for the needed developments. In this new scheme, the regional and national networks become the pillars of the global initiative, organizing clusters and becoming responsible for the processing, preparation and distribution of datasets that users will be able to access real time and with a machine-to-machine tool, obtaining always the most updated collection possible but keeping high standardization and common data policy. This will also lead to an increase of the FAIRness (Findability, Accessibility, Interoperability and Reusability) of the FLUXNET data that would ensure a larger impact of the unique data produced and a proper data management and traceability.

## 1 Introduction

25  The FLUXNET network is a self-organized network of eddy covariance sites managed by scientists that share data, ideas and competences across the globe (Baldocchi et al. 2001). The eddy covariance technique (EC) (Aubinet et al. 2012) allows a direct and not-destructive measurement of greenhouse gases (GHGs) and energy exchange between surface and atmosphere at ecosystem scale (500m to 1km around the measurement point) and typically half-hourly time resolution.

Since the first examples of year-long measurements (e.g. Black et al. 1996, Valentini et al. 1996), the use of EC data became more and more common not only to study single ecosystems from an ecological and physiological point of view (e.g. Reichstein et al. 2007, Law et al. 2002, Mahecha et al 2010, Luyssaert et al. 2007, Besnard et al. 2018) but also as ground observations in modelling development and validation and remote sensing applications (e.g. Bonan et al. 2011, Friend et al. 2007, Williams et al. 2009, Balzarolo et al. 2014, Jung et al. 2020). The large range of possible applications and the wide interest in these measurements, led first to the creation of regional and continental networks such CarboEurope (Dolman et al. 2006) and AmeriFlux (Novick et al. 2018) (followed by other continents for example with AsiaFlux, OzFlux, LBA and ChinaFlux, see Yamamoto et al., 2005, Beringer et al., 2016, Restrepo-Coupe et al., 2013 Yu et al., 2006) and then to the organization of the FLUXNET network-of-networks where all the regional networks contribute with a variable number of sites and years of data.

In the context of FLUXNET there have been different initiatives to facilitate discussion and cooperation across networks with specific conferences and meetings (starting in 1995, see Baldocchi et al. 1996) and the preparation of FLUXNET synthesis data collections with the aim to make the data available to wider communities. The main FLUXNET collections were produced in 2001 (Marconi dataset, Falge et al. 2005), 2007 (LaThuille dataset) and 2016 (FLUXNET2015 dataset, Pastorello et al. 2020), including an always larger number of sites-years (97 in Marconi, 965 in LaThuile and more than 1500 in FLUXNET2015) and providing standardized data ready for a large range of heterogeneous applications. These collections were needed because each regional network applies its own processing and formatting scheme (including different variable names and units) and this prevents an easy use of data across sites in different continents. In the last years AmeriFlux and the European networks worked toward a standardization that also highlighted the uncertainty introduced by the data processing (Pastorello et al. 2020) but this still not sufficient to replace global initiatives. However, the preparation of a FLUXNET collection requires a large effort that involves data collection, data policy agreement, common data quality controls, feedbacks with the site owners for corrections, processing and finally preparation of the products and their distribution, including the maintenance of the web-services for the data distribution, users tracking, updates of information etc.. All this considering that FLUXNET per se is not a funded initiative, there are no structural funds to maintain its operation and the synthesis dataset were created on initiatives of single groups often in the context of specific research projects. This is why 6 and 9 years passed between one FLUXNET synthesis collection and the following one.

The heterogeneity across regional networks is however something difficult to avoid. These networks are in fact based on general goals and scientific aims that can be different and can require specific design and processing. For example, the NEON network was planned using a hierarchical system to represent different ecoregions (Schimel et al. 2007) and the sites are highly standardized in terms of setup. Also in ICOS (Integrated Carbon Observation System) the stations are highly standardized but the design is driven by the single country decisions and priorities. In AmeriFlux, instead, an open participation is possible and everybody can register their sites in the network, without an overall design or standardization of the towers setup but allowing diversity and bringing under the same network sites designed for specific and heterogeneous research projects. In addition, single sites can be linked to other national or regional initiatives that could impose specific ways to prepare and distribute the

data collected. Finally, but often one of the most important aspects, there are different views, sensitivities and readiness respect to the data sharing and data use policies, often linked to the need of visibility (of both the single sites and the Regional networks)

that ensure proper funding to sustain the activities. These are key aspects, fully justified and difficult to change at global level in a short or medium period, which therefore need to be considered in a re-organization of the FLUXNET network structure

## 2 New needs and the role of FLUXNET

The need of ground observation data is increasing continuously and there are new examples of modelling and synthesis applications that require (or would require) direct measurements updated frequently. One example of such activities is the

FLUXCOM initiative (Jung et al. 2020), where satellite and meteorological spatialized data are used as input in a machine-learning (ML) ensemble to predict Net Ecosystem Exchange, Gross Primary Production, Ecosystem respiration and other energy fluxes at continental and global scale. These data represent often a link between the observations in FLUXNET and the large scale modelling initiatives. The ML algorithms need observations for their parameterization and the FLUXNET data have been successfully used in the training (e.g. Tramontana et al. 2016). Although the relations between drivers and fluxes

can be "learned" by the ML also using past data, the availability of new stations is crucial to improve the quality of the predictions and reduce their uncertainty. This is particularly relevant if new data cover under-sampled areas (Papale et al. 2015), extreme climatic events (Mahecha et al. 2017, van der Horst et al. 2019), different land management practices, and in general the effect of the climate pressure on ecosystems (Anderegg et al. 2020). An annual production of these bottom-up empirically upscaled estimations could for example be used as additional input in the Global Carbon Project

(www.globalcarbonproject.org) annual report (e.g. see Friedlingstein et al. 2019) on the carbon balance of the globe, where currently the FLUXNET data are in general not sufficiently used. The provision of a standard, continuous and global dataset of surface-atmosphere exchanges of GHGs is also a fundamental step to include the eddy covariance fluxes in the list of the Essential Climate Variables (ECV) define by GCOS for the empirical observation of processes related to climate change (Bojinski et al. 2014).

The same is valid for the remote sensing community that needs ground validation data frequently and with high quality standards, like in case of the Ground-Based Observations for Validation (GBOV) of Copernicus Global Land Products (https://land.copernicus.eu/global/gbov/home/) or the CEOS Land Product Validation (LPV) subgroup (https://lpvs.gsfc.nasa.gov/) that already cite FLUXNET as potential source of data but currently can not find a valid contribution because the data do not overlap in time with the most recent sensors (e.g. the Sentinel constellation).

Remote sensing community is also developing new tools that requires almost real time data (or with minimal delay) for the validation of their products that can also be of interest for the FLUXNET community. An example is the ECOSTRESS initiative for the evapotranspiration estimation where FLUXNET data have been already used (Fisher et al. 2020) but additional missions requiring a set of rapidly and directly available flux data will probably appear in the near future (e.g. Sun induced fluorescence or radar based products on soil moisture and canopy structure).Finally, there is a set of potential new fields and

applications that today are only partially using the FLUXNET measurements but would benefit from a more strong interaction with the eddy covariance community. These include, for example, the near-term ecological forecasting (Dietze et al. 2018), the use of FLUXENT data in weather forecast models (Boussetta et al, 2013) or the near-real time monitoring of agriculture. If we want to have the FLUXNET data more used and integrated with other scientific disciplines, also to start new cross-disciplines collaborations based on recent or even near real time data, we need to change the way in which the data are shared

in order to make their use more easy and suitable for new applications. In particular, we need to work to ensure fast updates of the collection and easy and direct machine-to-machine data access and data use capabilities, with a clear and easy to apply data use policy. Unfortunately we are not yet there and the use of an updated and standardized set of data still requires and extra effort (and a set of competences) that only few users are able to afford. For example Fisher et al. (2020) in their paper present very clearly the list of issues to address to create a usable collection, that span from a largely heterogeneous data format

(more than a dozen), processing level, collection mechanism to the need of an additional reformatting, processing and QAQC before the data use.

The characteristics of a dataset to ensure a machine findable and readable format and a clear rule for its use have been described by the FAIR principles (Wilkinson et al., 2016) and a new scheme should move in this direction (e.g. Collins et al. 2018). In particular, following the FAIR principles, the FLUXNET data should be easy to find (Findable) through common metadata

searchable by a tool; easy to access (Accessible) also through a machine-to-machine system and with a common and clear data use policy; processed in the same way and distributed in the same format in order to simplify the merging and synthesis (Interoperable); and clearly identified and permanently referenced in order to allow multiple uses and reproducibility of the studies and results (Reusable). All this, keeping the system robust and sustainable and for this reason not dependent on the capabilities and resources of a single network or group (as it has been until now).

The FLUXNET members would also benefit from a system able to process, standardize and distribute their data rapidly and in a clear and traceable way. The site teams would obtain a set of products as output of the centralized processing, that in some cases could be difficult and time and resources consuming to apply individually. In addition, and more important in my opinion, a FLUXNET network with these characteristics would provide new opportunities to the FLUXNET members for collaboration and joint activities, facilitating synthesis studies at continental and global scales. For example, the ICOS community promptly

prepared and shared a collection of in situ measurements from 52 sites in Europe (www.icos-cp.eu) that are used to analyse the effect of the 2018 European drought (e.g. Graf et al. 2020, Fu et al. 2020) on terrestrial ecosystems. This fast data release however was possible only thanks to an extra effort for the data processing by ICOS (in addition to the effort by the site teams to collect and share the data) and it is difficult to imagine this as standard way to proceed in future and globally. In fact, ICOS was created and funded as Research Infrastructure designed to sustain an organized observation network with prompt data

delivery but this is not common across all the regional networks that compose FLUXNET.

## 3 A new FLUXNET organization

In order to answer the new needs and opportunities described above, a new FLUXNET organization is necessary, that should start from the experience and development achieved and take into consideration the complexity of the system and peculiarities of all the participants. The solution should involve all the regional networks participating in order to increase the robustness and sustainability and, at the same time, keep their autonomy and internal flexibility needed to answer additional specific research questions, respect the organizational and political structures governing them and answer specific needs in terms of data processing, format and sharing.

For this reason, a new FLUXNET organization should be based on an agreement among the different regional networks, in order to ensure redundancy of competences particularly important in case of limitations in the resources. In the proposed scheme, the networks are grouped in FLUXNET clusters that agree to share data following a common procedure when the participating networks and the single sites are ready, interested or available to share (Figure 1).

With this organization, the FLUXNET clusters become the pillars of the FLUXNET system, coordinating the participation and data sharing in FLUXNET by different national and regional networks. In order to ensure the needed standardization in terms of processing, format, accessibility and data policy, the FLUXNET clusters must agree to prepare and maintain a specific database structure (the "FLUXNET baskets" in Figure 1) where a common and agreed data product (including all the needed metadata and versioning information) are loaded and made available. The main change respect to the current system is in the role of the Regional network databases and processing centres that would need to organize and run the cluster (Table 1). For the sites instead the system remains similar to the current organization (Figure 2), with the addition that it is not needed to organize double submissions of the same data (to the Regional network and for FLUXNET synthesis) but it is sufficient to decide when, for a given dataset, it is time to share in FLUXNET. In fact the Regional networks can continue to distribute data according to their specific data policy and move to the FLUXNET cluster only the dataset that can be shared under the common open data policy.

The FLUXNET product creation requires also that all the participating networks agree on the characteristics (for example minimal requirements about the variables, standard processing to apply, (meta)data format, common data policy, mechanism for data access etc.) and contribute to the development. However, we do not have to start from scratch: in the last years, for the preparation of the FLUXNET collections, standards have been already defined and implemented also at regional level (e.g. AmeriFlux, the European Database and ICOS produce already the same output). These include format, units, processing schemes and codes that are openly accessible, like in the case of the ONEFlux suite (Pastorello et al. 2019 and 2020).

Clearly the methods, standards and the needs evolve in time and for this reason it is important to discuss and agree on a plan and strategy to coordinate the efforts and define the common set of rules to apply in the FLUXNET clusters. FLUXNET worked well as bottom up initiative, community driven and without rigid and formal governing bodies, allowing people to participate, propose and use the FLUXNET organization in a democratic way. To keep this spirit, a light coordination

committee constituted by Regional networks and FLUXNET clusters representatives that work directly on data processing could serve as tool for the process governance in the definition of the new standards to apply and new products to introduce.

It is also important to define a strategy to evaluate and decide on implementation of changes or additions to the standards. In general, there is no reason to change established methods and formats if not motivated since this has an impact on the users that have to adapt their tools (in particular users interested to continuous data uses). For the processing the requirements could be, as in the last FLUXNET releases, that the processing tools should be at least 1) published in peer-review journals, 2) available to be easily applied to large and heterogeneous dataset, 3) with the implementation codes open source and 4) different

enough from what is already implemented to justify their addition to the processing flow (it is crucial to find the right balance between completeness and usability, too many options lead can lead to confusion).

The regional and national networks and single sites that are part of a FLUXNET cluster can continue to keep their specific databases and interfaces if needed (the Data portals in Figure 1) to distribute their data. This could be needed in case of different formats (e.g. when linked to other observation networks with different standards) or in case of different processing (e.g.

additional variables calculated centrally from raw data, or products of regionally specific processing tools). It should be noted that standard processing has the advantage of making all the data more comparable but at the same time it is possible that in specific conditions or sites it fails and an ad hoc specific processing is needed and results could be shared in the network Data portals. Differences in the data policies applied to specific sites or specific portions of the database can also be handled through regional data portals that can define licenses different respect to the common used in FLUXNET. Then, when a dataset become

ready to be shared in the FLUXNET system, it is processed also following the agreed FLUXNET standard and loaded in the FLUXNET basket.

The FLUXNET collection is then not any more a large dataset stored in one location but a set of sub-collections stored in the FLUXNET baskets of the different FLUXNET clusters and accessible visiting all of them to get the last version available. The access can be implemented through a common query system (the FLUXNET shuttle in Figure 1) that points automatically to

the different FLUXNET baskets and, using standardized metadata that include versioning information, gets the last version of the FLUXNET cluster collections to create an updated FLUXNET collection for the user. In this way, each single user could create at any time (on demand) a collection that is built using the most recent data provided by the FLUXNET network, allowing applications that requires updated collections. At the same time, the system gives the possibility to promptly correct possible errors if needed and to include continuously new sites as soon as they are ready to share, making FLUXNET even

more inclusive. In order to help scheduling of the work of the teams responsible of the sites, fixed "FLUXNET shuttle" runs can be scheduled for the main operational activities, e.g. before a FLUXCOM training or periodically when satellite products validation tasks are scheduled.

Clearly one of the requisites to have the FLUXNET shuttle working correctly and the users able to use the data is a common and clear data policy. The FLUXNET clusters must agree on a common data licence that should simplify and promote the use

of the data. With the aim to have FLUXNET used and promoted by different communities, standard data licenses should be considered because common across disciplines and for this reason well know. Currently most of the monitoring networks are

moving to the Creative Common CC-BY 4 license (https://creativecommons.org/licenses/by/4.0/) that ensure attribution and promote data use. All this, however, must also considered the need of recognition and advantages for the scientists working at the sites that are discussed below.

**4 Advantages and risks of the proposed new organization**

The proposed FLUXNET scheme would have a number of advantages. First, the users will not have to wait for releases of datasets every 5 or 10 years but can get the most updated version of the shared data in real time. This would stimulate the use of data by scientific communities that need recent measurements (e.g. in early detection of anomalies). The data would increase also their level of FAIRness, improving their Findability through the use of standard metadata across the FLUXNET clusters,

their Accessibility through a common open data policy and a single tool to retrieve all the data (the FLUXNET shuttle), their Interoperability thanks to the standardization. With a system that creates a new (and potentially different) collection at every user's request, it is crucial to clearly identify the data included (and the versions) also to ensure reproducibility of the results. This is achievable through a specific persistent identifier (PID) that users should always report and that will improve the data Reusability in case of studies reproduction and verification.

In terms of robustness, sustainability and flexibility, the proposed system would also substantially improve the current situation thanks to the overlap of data processing capacities and responsibilities among the FLUXNET clusters. In fact, sharing of workload will stimulate collaboration across networks and promote interchangeability of roles since each FLUXNET cluster could process the data of another cluster if needed. This crucial aspect is missing today; if for example one network or FLUXNET cluster has difficulties in a certain period (lack of funding, key people moving etc.), the other FLUXNET clusters

can support the common processing so that the network with difficulties could dedicate the resources only to internal discussion with the sites and data collection. This could be particularly relevant in case a big changes in the processing scheme (that will inevitably happen) and that will require a massive data reprocessing. In this case, a mutual support of the FLUXNET clusters or also an investment on a common and shared computing resource for the standard processing would help the sustainability for all the networks.

The capacity to process the data following the same standard method and the alignment in terms of code versions can be periodically tested though a verification system similar to a "round-robin test" where all the clusters will have to process the same set of data with the standard procedure and results are compared. All this keeping the full flexibility of each single network to decide what to share and when in FLUXNET and the possibility to distribute different formats and versions through their Data portals.

It is however important to analyse the concerns that a new FLUXNET organization like the one here proposed could raise. In particular, there is the risk of losing the control of the data (who accessed, where they are used etc.) and this is directly linked to a crucial aspect: the visibility of the people. The large amount of work and investment done by single stations and networks participating to FLUXNET must be fully recognized and should have an effect on the funding to continue the work and data

provision and on the career of the people involved. The contribution of data to FLUXNET is in most cases on voluntary bases
so the proposed system would not force participation. It is however important to try to get as many people and networks as possible engaged and the analysis of the benefits that data sharing can brings is the natural step to take a decision. Although this has been discussed in different frameworks (e.g. Papale et al. 2012) and studies demonstrated that people sharing data get more recognition due to the collaborations established (Bond-Lamberty, 2018; Dai et al., 2018), it is out of scope here to enter in the details on the benefits and convenience of data sharing.

What a reorganized and truly international FLUXNET system can do is to ensure a full traceability of data access and data uses, to allow each data owner to have an exact quantification of the use of the data shared. From a technical point of view, the compilation of a list of downloads per site it is something that can be easily implemented using the FLUXNET Shuttle and can provide important information about the use of the data. However, this is not enough: it would be important to have in all the papers that use these data the citation of the datasets so that the impact and usefulness of each single site can be quantified

and recognized. This would require the help of the journals that should request, during the review, to clearly cite the DOI or PID of the dataset used, and this should not be affected by the limitation in the number of citations often imposed. In this way it would be possible to evaluate and show the importance of the data collected and distributed by FLUXNET and which are the communities using them. Finally, a new and more robust, sustainable and fast organization could stimulate the interaction with the private sector that is currently missing (except for the instrument manufactures). Private users interested not only to

use but also to contribute to the measurements could increase the FLUXNET visibility and attract the needed resources to growth and strength the link with the stakeholders (Marino 2020).

## 5 Moving toward the implementation

A change in the FLUXNET organization, although based on the existing capacities and experiences of the site teams, regional networks and past collections leaders, can only be gradual with a transition phase that must allow all the interested groups to

adapt and organize their role and work. During this transition phase, it is important to keep present the overall aim and final structure but the activity can start from few initial groups that, for historical reasons or contingent situations, are ready to start prototyping the system. For example, ICOS and AmeriFlux are already distributing data processed using the same software (ONEFlux, Pastorello et al. 2020) in their respective portals (ICOS: https://meta.icos-cp.eu/collections/ueb_7FcyEcbG6y9-UGo5HUqV, AmeriFlux: https://oneflux-beta.ameriflux.lbl.gov/). The access is still individual and the policy different but it

is a first step in the direction of a distributed preparation and access to a common product.

During the transition phase it is important that FLUXNET remains inclusive, giving the possibility to everybody to get involved and have data processed and shared, without the risk to feel isolated or excluded. This can be ensured by a cross-networks support system, where clusters ready to process and distribute can temporary offer to do the activities for other networks or individual unaffiliated sites with, and here it is a difference respect to the current system, the agreement that in parallel all the

networks work in the direction of the establishment of a reference FLUXNET cluster. It is also clear that a single Regional

network could act as FLUXNET cluster autonomously, this is possible and it is only a matter of optimization in the use of resources.

It is also needed to discuss and agree on all the technical details, which can start from the experiences already done in the context of the FAIR principles applications and development and prototyping of specific tools (e.g. see https://envri.eu/home-envri-fair/). The choices regarding the organization of FLUXNET clusters, the technology to use, the timeline for implementation and all the other technical details need a general discussion where all the regional networks should be involved independently of their readiness in the actual implementation

## .6 Conclusions

The main differences between the current FLUXNET organization and the new proposed structure are the shared workload and overlap of competences among a number of organizations (FLUXNET clusters) that can ensure the needed robustness and the real time distribution of new data available. All this without scarifying the visibility and role of the Regional networks that remain crucial for their role of organization, support, guidance and scientific development linked to the local networks. The main benefits would be 1) an increase of robustness of the global network thanks to the sharing of workload and responsibilities, 2) a strength of the collaborations among networks and colleagues across the world and 3) an increase of visibility thanks to the continuous availability of updated products that can lead to more users and resources. There are clearly also risks like in all changes that however can be handled with a smooth transition phase and a real spirit collaboration (Table 2). The solution is also scalable once implemented, giving the possibility to include new measurements (e.g. new GHGs like $CH_4$ or $N_2O$, see Knox et al. 2019, Nemitz et al. 2018) or new processing also starting from raw data. In fact the development of new tools by a FLUXNET cluster, already designed to be generally applicable, can be made available to all the others easily and without duplication the efforts. The proposed scheme would also move FLUXNET in the direction that was already defined 20 years ago, developing a collaborative, self-organized and bottom up network, able to answer to new requests thanks to the continuous updates. This can works also as example for similar distributed observational networks that could benefits from the experience done in reorganizing FLUXNET. The evolution of the regional networks toward more organized and stable infrastructures, the large number of eddy covariance people that are now sharing data and collaborating in FLUXNET and the new spirit of collaboration among regional networks, are solid bases to do this step.

*Competing interests:* the author declares that they have no conflict of interest.

*Acknowledgements:* the author thanks all the colleagues and friends that shared with him ideas and comments on the development of FLUXNET and thanks the whole FLUXNET community for the very constructive and open spirit that helped

to build a so nice bottom-up coalition. Thanks also to the reviewers and the editor for the comments and suggestions needed
to improve the clearness and completeness of the manuscript.

*Financial support:* the author thanks the support of the RINGO (Grant Agreement 730944) and ENVRIFAIR (Grant Agreement 824068) H2020 European projects for the development of a new and more integrated scheme of FLUXNET and the E-SHAPE (Grant Agreement 820852) H2020 European project to support a first pilot study on the operational use of FLUXNET data.

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

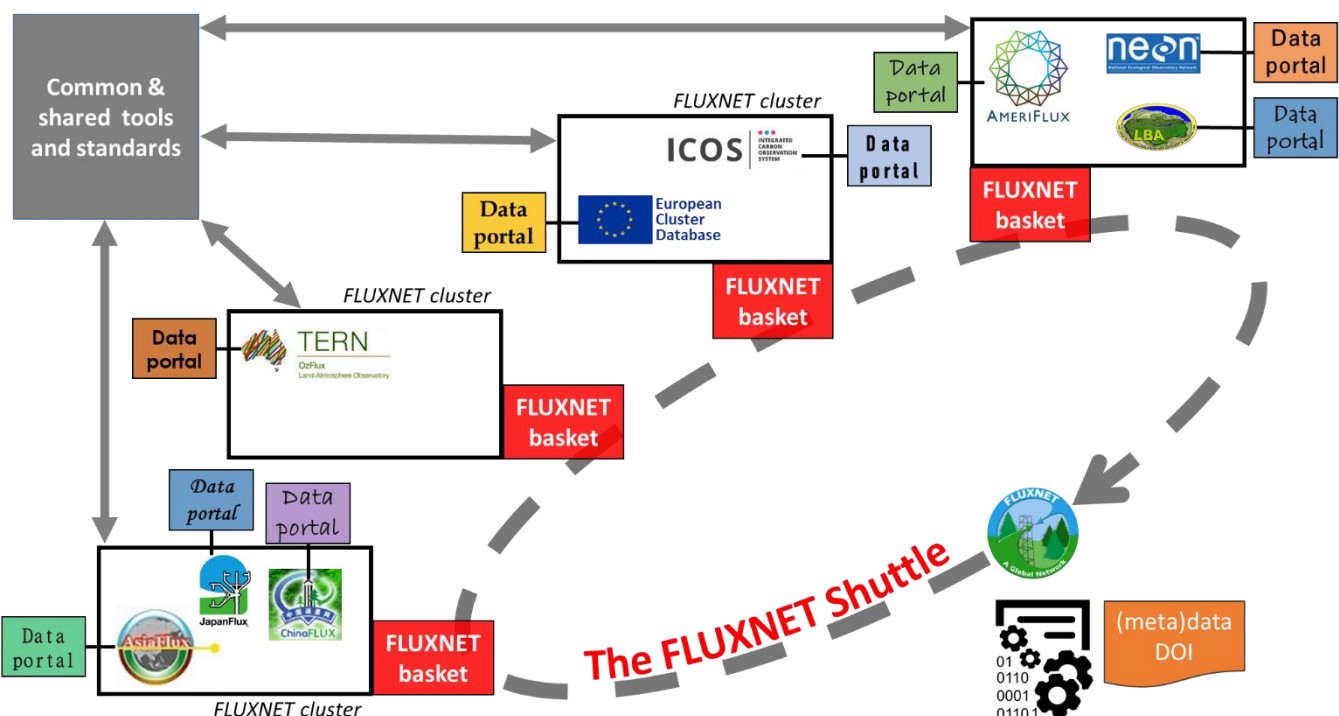

**Figure 1: scheme of the proposed new organization of FLUXNET for data collections preparation (see text). "Data portal" boxes represent the regional/national network databases, all potentially different in terms of data processing, format and data policy. The black boxes grouping regional/national networks are the "FLUXNET cluster", the framework under which a set of national networks coordinate their participation in FLUXNET and where a common processing is applied. "FLUXNET basket" red squares are the database sections for FLUXNET data to share, where common format of data and metadata are loaded whenever ready and distributed under the same and common data policy. "FLUXNET Shuttle" is the tool to access the data across the FLUXNET clusters that is run on-demand by the users and provide a dataset (including metadata) and a PID or DOI for the exact citation and reconstruction of the dataset used.**



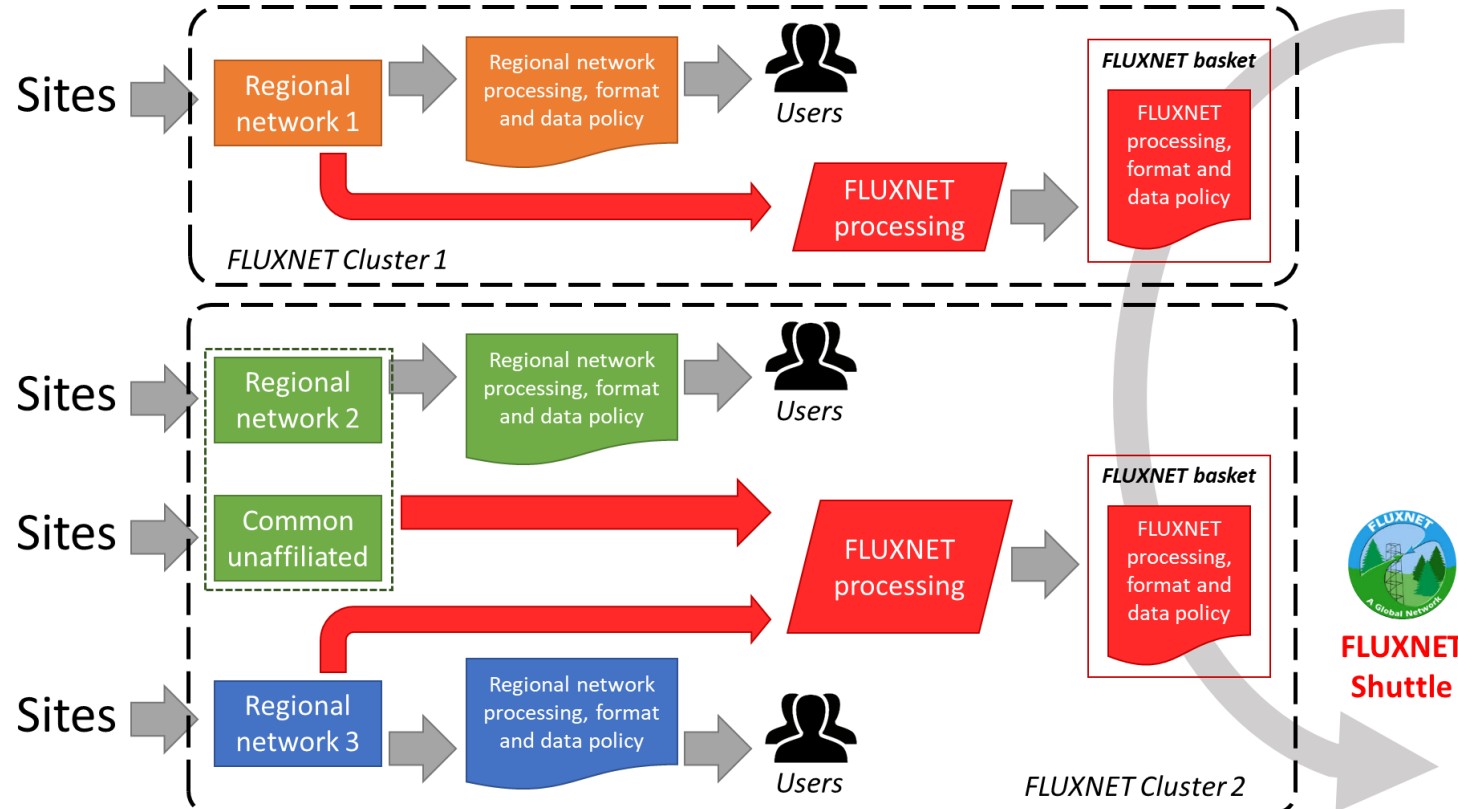

**Figure 2: data flow from the sites to the FLUXNET Shuttle. The sites submit the data to the Regional networks where they are associated or, may be for a temporary period, to a common system for unaffiliated sites that is managed by one of the Regional networks (in the figure the Regional network 2). Each Regional network can organize its own data processing, data policy and data distribution system. Part of the data are then also processed using the standard FLUXNET processing and then shared in the FLUXNET basket where the FLUXNET shuttle can collect for the user on request. The data shared in the FLUXNET shuttle are defined by the data owner. Note that the clusters can be also composed by a single Regional network (like for Regional network 1 in the figure) if the resources are sufficient to maintain it.**


| Component | Action | Current system | Proposed system |
|---|---|---|---|
| **Sites** | Data submission date | *After a call for synthesis, respecting a deadline* | *As soon as ready or interested* |
| | Data submission method | *To the people initiating the synthesis* | *To the Regional network (temporary if needed to a common platform)* |
| | Data policy | *Two or three options to select* | *One policy, common for everybody* |
| **Regional Networks** | Data collection | *Some networks collect from their sites* | *Data collection for all the sites participating* |
| | Data processing | *none* | *Contribute to the FLUXNET Cluster* |
| | Data storage | *Original data* | *Original data and FLUXNET products* |
| | Data distribution | *Original data* | *Original data and FLUXNET products through the FLUXNET Cluster* |
| **FLUXNET Cluster** | Data collection | *not existing* | *none* |
| | Data processing | *not existing* | *Apply standard FLUXNET data processing* |
| | Data storage | *not existing* | *FLUXNET products* |
| | Data distribution | *not existing* | *Organize and maintain FLUXNET basked for the sharing through the shuttle* |
| **FLUXNET Synthesis team** | Data collection | *Collect from all the sites and Regional Networks* | *Collaborate through the Regional Networks and FLUXNET Clusters* |
| | Data processing | *Apply standard FLUXNET data processing* | *Collaborate through the Regional Networks and FLUXNET Clusters* |
| | Data storage | *FLUXNET products* | *Collaborate through the Regional Networks and FLUXNET Clusters* |
| | Data distribution | *Organize and maintain a FLUXNET server for distribution* | *Collaborate through the Regional Networks and FLUXNET Clusters* |

**Table 1: main changes for the different actors between the current FLUXNET synthesis system and the one proposed in this paper. The FLUXNET Cluster does not exist in the current organization and it is the key new component proposed.**

| Strengths | Weaknesses | |
|---|---|---|
| | **Point** | **Corrective action** |
| Distributed workload that ensures sustainability and robustness | Investment done until now is not used | The competences will migrate in the new system |
| Continuous updates of the collection | Risk that the data policy is not followed | The new system make all more engaged to ensure proper data citation |
| Easy data access and clear policy | Feeling that the data control is lost | The FLUXNET Shuttle will have to register all downloads and provide PIDs |
| Increase visibility of the Reginal Networks and engagement of the regional communities | Sites could be not ready/interested to adopt the standard open policy | The Regional networks can continue to distribute the data under their policies |
| **Opportunities** | **Threats** | |
| | **Point** | **Corrective action** |
| Attract more users and interests | Only few Reginal Networks able to organize this | Other Regional Networks could help |
| Stimulate participation also from less represented areas | Distributed processing could affect standardization | Periodic tests using a "Round robin" method |
| Increase visibility and international collaboration | Readiness of the Regional networks not homogeneous | Transition phase where a general FLUXNET Cluster is also active |
| Get more stable funding from other organized users | | |

**Table 2: SWOT (Strengths, Weaknesses, Opportunities, Threats) analysis of the new proposed system. For the Weaknesses and Threats possible corrective actions are also reported.**