# Peer review of "Ideas and perspectives: enhancing the impact of the FLUXNET network of eddy covariance sites"

_Biogeosciences, 2020_

## Referee Comment (RC1) · Joshua Fisher (Referee) · 23 Jun 2020

To be transparent with my biases, I am a user of eddy covariance data, not a producer. So, I'm going to be hard pressed to find anything against this paper, as it's essentially designed to help someone like me. I can certainly imagine there being pushback from individual data producers and networks, which I imagine justify why this proposed standardization hasn't come to fruition thus far. But, from my perspective, I can only really provide even more evidence and support for this paper. I suppose I could think of ways to make it even better for someone like me; but, this relative to the status quo is such a big improvement that I wouldn't want to make it less likely to occur by adding on any more bells and whistles. I hate to be a reviewer without a lot of complaints, and disparaging and unfounded insults—I mean, unleashing one's own personal frustrations onto unsuspecting authors is therapeutic, right? (Totally kidding…). I did find one spelling error though, so clearly the author doesn't know what he's doing (totally kidding again!).

- "Here a proposal for a new organization where regional and national networks become the pillars of the global initiative, organizing clusters and becoming responsible for the processing and preparation of datasets…"
  - Isn't this already the *old* organization? I know what you're trying to say here, but it doesn't come across clearly (essentially, those pillars aren't always under the same ceiling, and so it takes some moving of those pillars around in the syntheses, which is time consuming).
  - Is it really a new proposal? Hasn't this been proposed for years?
- Certainly, the author (and his acknowledged though not co-authored colleagues) have thought this through pretty thoroughly. But, why aren't there any other co-authors? Wouldn't it be more convincing if there were at least one co-author from each of the networks? It's slightly ironic that the article is about collaboration when there are no co-authors.
- Overall, the paper is somewhat light on specific details of how everything would work. I would guess that people are mostly on-board in theory. But, the practical systems engineering could perhaps be flushed out a bit more. Perhaps an additional figure could be useful that would reflect this.
- Are there analog data networks that could be discussed for failures/successes?
- I wonder if FLUXCOM is the best example to justify the proposal. Adding new data to FLUXCOM at this point changes it very little, as far as I would expect, and it moves without real time eddy flux data based on globally gridded inputs. I guess the justification might be better if it were for FLUXCOM-like new initiatives; or, new members to FLUXCOM.
- I'm trying not to be biased, though I definitely am, but a good example for the remote sensing need was published in *Fisher et al.* [2020] for the ECOSTRESS mission, focused on evapotranspiration. Here, we needed as much current eddy covariance data as possible right away (i.e., data before launch would be far less useful!) to ensure that our ET data products were good enough to release. It was a beast to deal with all the different disparate data (as you obviously know)—more than a dozen data formats alone, let alone the interfaces to access the data! We mentioned some of those aspects in the Methods section of the paper, check it out. In the end we had an amazing >150 sites contribute data (with nearly as many co-authors, because, after all, the eddy flux data were all new

too). I suspect that this paper is why the Biogeosciences Editor here thought of me to ask to be a reviewer. Our validation work also followed on similar validation work done for the SMAP mission on soil moisture.

- We're going to continue to have new missions that would benefit greatly from the proposed global standardized network. More missions that deal directly with fluxes, as well as others that deal with other variables that are useful from FLUXNET sites such as soil moisture, canopy height, fluorescence, etc.
- It would be good to include a statement of justification for Continental clusters. (Also, continents are not always consistently defined across the world).
- L167. "wait releases of dataset" → "wait for releases of datasets"
- L188-193. Check out the little-known *Fisher and Fortmann* [2010], where we applied Elinor Ostrom's design principles for sharing of natural resources to shared data with an application to FLUXNET. Keys to successful data sharing discussed therein.

I hope to see this proposed organization a reality soon!

Josh Fisher

Fisher, J. B., and L. P. Fortmann (2010), Governing the data commons: Policy, practice, and the advancement of science, *Information & Management*, *47*, 237-245.

Fisher, J. B., B. Lee, A. J. Purdy, G. H. Halverson, M. B. Dohlen, K. Cawse-Nicholson, A. Wang, R. G. Anderson, B. Aragon, M. A. Arain, D. D. Baldocchi, J. M. Baker, H. Barral, C. J. Bernacchi, C. Bernhofer, S. C. Biraud, G. Bohrer, N. Brunsell, B. Cappelaere, S. Castro-Contreras, J. Chun, B. J. Conrad, E. Cremonese, J. Demarty, A. R. Desai, A. De Ligne, L. Foltýnová, M. L. Goulden, T. J. Griffis, T. Grünwald, M. S. Johnson, M. Kang, D. Kelbe, N. Kowalska, J.-H. Lim, I. Maïnassara, M. F. McCabe, J. E. C. Missik, B. P. Mohanty, C. E. Moore, L. Morillas, R. Morrison, J. W. Munger, G. Posse, A. D. Richardson, E. S. Russell, Y. Ryu, A. Sanchez-Azofeifa, M. Schmidt, E. Schwartz, I. Sharp, L. Šigut, Y. Tang, G. Hulley, M. Anderson, C. Hain, A. French, E. Wood, and S. Hook (2020), ECOSTRESS: NASA's Next Generation Mission to Measure Evapotranspiration From the International Space Station, *Water Resources Research*, *56*(4), 1-20.

---

## Author Comment (AC1) · 25 Jun 2020

*First let me thanks Joshua Fisher for the review and comments, that are useful and appreciated. I decided to give my answers and clarifications now, without waiting the end of the Discussion period, because I hope these could be useful also for other readers of the current version. My comments and answers are in blue italic*

1) Here a proposal for a new organization where regional and national networks become the pillars of the global initiative, organizing clusters and becoming responsible for the processing and preparation of datasets. . ." - Isn't this already the *old* organization? I know what you're trying to say here, but it doesn't come across clearly (essentially, those pillars aren't always under the same ceiling, and so it takes some

moving of those pillars around in the syntheses, which is time consuming).

*It is not, otherwise it would be already implemented... The main differences are in the 1) agreement among all the participant to push this forward and 2) role of the different groups with distributed responsibilities. I agree that looks a simple and easy concept, but practically it does not exist. I will try to clarify better in the new version the main problems and why the current organization does not work and highlight the differences respect to the current situation.*

- Is it really a new proposal? Hasn't this been proposed for years?

*It is not, otherwise it would be already implemented... The main differences are in the 1) agreement among all the participant to push this forward and 2) role of the different groups with distributed responsibilities. I agree that looks a simple and easy concept, but practically it does not exist. I will try to clarify better in the new version the main problems and why the current organization does not work and highlight the differences respect to the current situation.*

2) Certainly, the author (and his acknowledged though not co-authored colleagues) have thought this through pretty thoroughly. But, why aren't there any other co-authors? Wouldn't it be more convincing if there were at least one co-author from each of the networks? It's slightly ironic that the article is about collaboration when there are no coauthors.

*I get the point and agree on the fact that it is strange. In fact, however, the proposal on how to reorganize the structure is something I developed and then presented in some meeting in a very simplified and schematic way. What I think is needed to get it moving forward and being implemented is exactly something like this paper where the concept is presented with more details and open to comments thanks to the Biogeosciences review procedure. It is a global initiative, having only some people involved at this stage would have been even more dangerous I would say. So this is an "Idea and Perspective" from me, open to discussion and modification and then to technical discussion on the implementation (hopefully)*

3) Overall, the paper is somewhat light on specific details of how everything would work. I would guess that people are mostly on-board in theory. But, the practical systems engineering could perhaps be flushed out a bit more. Perhaps an additional figure could be useful that would reflect this.

*Good suggestion, thanks. The details on the metadata and query systems are too much technical in my opinion, but something explaining better the overall path from a PI perspective and interaction with the different components, including the data flow, could be important and useful. I will add it in the revision.*

4) Are there analog data networks that could be discussed for failures/successes?

*Not that I know with a similar structure. In my opinion FLUXNET is quite advanced and I'm always impressed when I realize this because for its nature (self-organized, not officially defined, loose structure) it is really advanced and working well (thanks to all the people that are actively participating, sharing, organizing and also using it). I would also say that this could be an example for other networks.*

5) I wonder if FLUXCOM is the best example to justify the proposal. Adding new data to FLUXCOM at this point changes it very little, as far as I would expect, and it moves without real time eddy flux data based on globally gridded inputs. I guess the justification might be better if it were for FLUXCOM-like new initiatives; or, new members to FLUXCOM.

*FLUXCOM is evolving and always open to new methods and improvements. In addition the climate-ecosystems interaction are changing with the more and more frequent climatic extremes and it is crucial to have new data to parameterize the machine learning techniques in these "new" scenarios. I will add in the revised manuscript a more detailed explanation about the importance of using measurements under climatic driven pressures (e.g. Anderegg et al. 2020 in Science).*

6) I'm trying not to be biased, though I definitely am, but a good example for the remote

sensing need was published in Fisher et al. [2020] for the ECOSTRESS mission, focused on evapotranspiration. Here, we needed as much current eddy covariance data as possible right away (i.e., data before launch would be far less useful!) to ensure that our ET data products were good enough to release. It was a beast to deal with all the different disparate data (as you obviously know)—more than a dozen data formats alone, let alone the interfaces to access the data! We mentioned some of those aspects in the Methods section of the paper, check it out. In the end we had an amazing >150 sites contribute data (with nearly as many co-authors, because, after all, the eddy flux data were all new too). I suspect that this paper is why the Biogeosciences Editor here thought of me to ask to be a reviewer. Our validation work also followed on similar validation work done for the SMAP mission on soil moisture.

*It is not a matter to be biased, your example is of great interest and I was looking for a publication like the one you cited in the manuscript preparation. I agree that it is a perfect example and I will definitely add it in the paper. The future of FLUXNET in my opinion is strongly linked to the collaboration in initiatives like ECOSTRESS, where getting the data should not be a "MENTALSTRESS" but something smooth and clear.*

7) We're going to continue to have new missions that would benefit greatly from the proposed global standardized network. More missions that deal directly with fluxes, as well as others that deal with other variables that are useful from FLUXNET sites such as soil moisture, canopy height, fluorescence, etc.

*I completely agree. I will try to incorporate the concept more clearly in the paper.*

8) It would be good to include a statement of justification for Continental clusters. (Also, continents are not always consistently defined across the world).

*I agree, may be "Continental" is not the right way to call them, also because in the system in theory each single network can organize its "FLUXNET Version" basket. I will revise this and the Figure to better explain that this is an option to save resources if needed or to support smaller networks that don't have the possibility to maintain the*

*system in place.*

9) L167. "wait releases of dataset" à "wait for releases of datasets"
*Thanks*

10) L188-193. Check out the little-known Fisher and Fortmann [2010], where we applied Elinor Ostrom's design principles for sharing of natural resources to shared data with an application to FLUXNET. Keys to successful data sharing discussed therein.
*Thanks for the suggestion, I will check it.*

11) I hope to see this proposed organization a reality soon!
*Me too... thanks for the support to the idea*

---

## Short Comment (SC1) · 26 Jul 2020

Papale summarizes the history of FLUXNET and related networks resulting in inter-mittent data products and the current status of mixed interoperability. Papale notes that fluxnet community operations are not standardized, or available in near real-time, or effectively shared creating a mixed-mosaic of data formats and user requirements that are difficult to navigate (see review by JB Fisher). Solutions are proposed to re-align flux community/individual efforts and standardization of data format and access, however, few details as to engineering structure and how these solutions could be im-plemented are provided. While flux data for CO2 is produced and consumed primarily by research communities, it is of high importance to commercial applications for GHG mitigation and verification of emission reduction claims now emerging in the private

sector. Without direct measurement of CO2 flux (e.g., forests, soil, agriculture) at appropriate scales, commercial applications cannot contribute to solutions based on the net removal of GHG's.

Adoption and commercialization of typical research instrumentation for eddy covariance will catalyze innovation and cost reduction as well as manufacture of turn-key eddy covariance systems that can be readily deployed anywhere on the planet. Regional networks of up to 1,000 nodes should be readily achievable at reasonable cost. In addition, interoperability should be seamless and draw on a standardized System of Systems architecture design accommodating high data volume, data security, third-party verification, and related concept of operations typical of the banking and related industries.

The vision proposed may seem bold and unachievable, however, we live in a time where real-time dynamic data characterizing the biosphere and anthropogenic impacts on the biosphere are crucial to the management of climate change now and for future generations. Off-the-shelf GHG analyzers, simple steel construction and fabrication of easily deployed single and multiple pole-based towers, and the ubiquitous presence of the internet and satellite telemetry are available and mature industries, poised for application-specific innovation such as proposed by Papale.

Importantly, private sector involvement may offer new sources of funding for vast eddy covariance observation nodes with the condition that all data remain freely available to the research community.

In summary, Papale proposes that a new "FLUXNET" emerges from a reorganization of existing fluxnet communities. This is a critical transition for FLUXNET, or a likely successor entity, to ensure continued growth and relevance to research and private sectors. The private sector should be actively involved in this transformation benefitting all stakeholders.

---

## Referee Comment (RC2) · Jason Beringer (Referee) · 29 Jul 2020

I am the current director of OzFlux and I bring this perspective but these comments are personal ones and not necessarily the views of the network or NCRIS TERN.

I wholeheartedly agree in principle with the philosophy of new approach to FLUXNET is terms of ensuring timely data, interoperability and to facilitate data use. We do need a new organisational structure to make sure that FLUXNET is sustainable into the future. In the past FLUXNET collections have only been possible due to the herculin efforts and passion of a few people and we are all extremely grateful for this but it is a big sacrifice that you and others have made.

In addition to the demand for real time data that you have outlined, there is also

an emerging area of ecological forecasting, for example, The EFI-NEON Research Coordination Network that is an NSF project to create a community of practice that builds capacity for ecological forecasting using NEON. There is also potential demand from data assimilation of flux tower data into short range weather forecast (e.g. https://doi.org/10.1175/MWR-D-19-0370.1). Also real-time agricultural monitoring should be possible. Finally, data can also be used to generate regional real time evapotranspiration estimates using a fusion of flux tower, remote sensing and modelling all of which have potential for use by the public.

With respect to the proposed re-organisation from a single large database into sub-collections. On the one hand this helps make the network more sustainable by delegating work and responsibility to continental clusters so that not all the work is being done by a small group. On the other hand this will require continental clusters to be functional, accessible and have open data sharing policies. I would be concerned that many clusters may not have capacity to do this or there may be a difference in data sharing between groups and individuals, such that individual sites that may want to contribute are unable because of the inability of the cluster to participate for technical, personnel or other reasons.

Following on from this I can see that there could be many, many sites (even some that were not part of FLUXNET2015) that want to contribute to FLUXNET but they are unable too because they don't have a functional continental cluster. It will be crucial to make sure there is a mechanism for them to contribute. The ONEFLUX processing code it not designed for this purpose. So I wonder if there could be an online processing tool that registered site users could upload their site data at what ever intervals they are able to do so?

I'm not entirely sure for the rationale and need to move away from a large centralised data base to Continental cluster collections? There has been a lot of effort gone into making the database and I'm not sure if it is broken in some way or has reached its capacity technically. It seems to add another layer of complexity to have Continental

cluster collections and a shuttle they queries each of them. It then relies to the ability of the clusters to maintain the data in real time in these clusters. Why not just have a continually updated big database where data is added at any time (daily or annually as it becomes available). You then rely on sites and clusters to push the data continually rather than calling and hassling for sites to submit data (current model). Users can query the data anytime too. We could build a set of tools that allow data to be accessed and queried in more sophisticated ways.

I like the idea of a persistent identifier (PID) or something similar. I would envisage that initially the processing would follow the FLUXNET2015 (i.e. ONEFLUX) pipeline and the PID would reflect that. But processing methods do evolve and a Fluxnet steering group could endorse any changes to the pipeline and periodically the PID would change to say FLUXNET2025 for example.

It is probably important to think about changes in processing and reprocessing the whole database. This may well happen in the future if we have a new pipeline and you would want to apply the new pipeline retrospectively across all the data I presume? This would be manageable under a single large database but may be difficult under Continental cluster collections.

As Papale acknowledges, it will be important full flexibility of each single network to decide what to share and when in FLUXNET and the possibility to distribute different formats and versions through their Data portals. I would envisage that in OzFlux, one could have rapid processing of near real time fluxes using the FLUXNET pipeline on a daily basis and this would have PID that differentiated these data streams as 'beta' datastreams or something. These data can be used for applications that require real time data but the data comes with the caveats of not being quality controlled by site investigators. The data should be fit for the purpose that it is being applied. As Ray Leuning would tell us, "know thy site" and investigators would still produce the finalised data sets with human skill and site knowledge. So we can have our cake and eat it too.

So I think there are potential pros and cons and maybe the best way is to get a working group together and flesh out an operational model for the future. I am very keen and you can count us in.

---

## Author Comment (AC2) · 30 Jul 2020

*Also in this case let me thanks Jason Beringer for the review and comments that are useful and appreciated. Here below my comments and answers on the specific points. My comments and answers are in blue italic*

In addition to the demand for real time data that you have outlined, there is also an emerging area of ecological forecasting, for example, The EFI-NEON Research Coordination Network that is an NSF project to create a community of practice that builds capacity for ecological forecasting using NEON. There is also potential demand from data assimilation of flux tower data into short range weather forecast (e.g. https://doi.org/10.1175/MWR-D-19-0370.1). Also real-time agricultural monitoring

should be possible. Finally, data can also be used to generate regional real time evapotranspiration estimates using a fusion of flux tower, remote sensing and modelling all of which have potential for use by the public.

*Thanks for the additional suggestions on the uses; I will improve this section because I agree that a larger set of potential user communities would work in the direction of a stronger and more central FLUXNET*

With respect to the proposed re-organisation from a single large database into sub collections.   On the one hand this helps make the network more sustainable by delegating work and responsibility to continental clusters so that not all the work is being done by a small group. On the other hand this will require continental clusters to be functional, accessible and have open data sharing policies. I would be concerned that many clusters may not have capacity to do this or there may be a difference in data sharing between groups and individuals, such that individual sites that may want to contribute are unable because of the inability of the cluster to participate for technical, personnel or other reasons.

*This is a good point and helps me to understands that my test was not clear enough (and so I will work on it).  The Continental clusters are a way to become more interoperable, with the specific meaning that the competences, capacity and resources to do the same type of activity are shared by different groups. This would give to the system the needed robustness thanks to redundancy.*

*However Jason Beringer is right saying that the "Continental clusters" should be able to do this and this is not always the case, in particular in a starting phase. Here there are few possible solutions/clarifications that I would better clarify in the new version:*

*- Continental is probably a wrong term. A cluster can be also just a single county if they have the capability or can be a joint initiative across continents (Asia+Australia just to make an example). In other words it is a scalable concept: the main point is to have number a groups able to produce the FLUXNET agreed products and care about their access.*

*- We can also have, for a period, an additional generic "FLUXNET basket" managed by a group that has resources/capacity, where single sites or clusters not ready can be processed and shared. But the aim must be to move in the direction that these groups become first able to use shared resources to do the processing (e.g. a cluster hosted in Europe to process themselves the data) and then to organize a Continental cluster that can then be a catalyst for regional networks development.*

*- On the data policy: it is not needed that the continental cluster adopts the common open policy in general, only that is able to share under this policy data that the PIs agree to share and document them with the needed metadata to be FAIR.*

Following on from this I can see that there could be many, many sites (even some that were not part of FLUXNET2015) that want to contribute to FLUXNET but they are unable too because they don't have a functional continental cluster. It will be crucial to make sure there is a mechanism for them to contribute. The ONEFLUX processing code it not designed for this purpose. So I wonder if there could be an online processing tool that registered site users could upload their site data at what ever intervals they are able to do so?

*This is also important, true and somehow connected (and partially answered) above. In a "Research Infrastructure" view, the online tool does not solve the problem, rather make it more complex because someone then has to ensure that the processed data are according to the standard, add the metadata and share them. All this with high risk of losing the traceability.*

*In my view the solution is to 1) stimulate and support the creation of 3-4 Continental Clusters that can offer the service to all the sites, also independently respect to the physical inclusion in the geographic continent; 2) prepare and maintain for the moment a generic cluster to host sites or small groups where a Continental cluster is not ready. It is a medium term process, having intermediate steps is needed.*

I'm not entirely sure for the rationale and need to move away from a large centralised data base to Continental cluster collections? There has been a lot of effort gone into making the database and I'm not sure if it is broken in some way or has reached its capacity technically. It seems to add another layer of complexity to have Continental cluster collections and a shuttle they queries each of them. It then relies to the ability of the clusters to maintain the data in real time in these clusters. Why not just have a continually updated big database where data is added at any time (daily or annually as it becomes available). You then rely on sites and clusters to push the data continually rather than calling and hassling for sites to submit data (current model). Users can query the data anytime too. We could build a set of tools that allow data to be accessed and queried in more sophisticated ways.

*In theory, a large database could also work. But it would require a constant and substantial funding to do the call for data submission, quality check and processing. Even if the centralized database is only for distribution, it would still require funding and support (by all the networks, with all the complications due to trans-continent funding) for the curation and maintenance. The database, if we want to have PID, citations counts, good metadata etc. is something substantially different respect to a "simple" repository.*

*However, the main reasons why I think that the centralised database is not the robust solution in the medium term are: 1) it doesn't help to build the distributed competences needed for the robustness and 2) as said also in the last comment, regional and continental clusters wants also their cake and being more connected to FLUXNET (from collection to production and distribution) would help.*

*The submission of data to a central database does not work automatically, believe me, it requires a lot of efforts as demonstrated by the activities in LaThuile and FLUXNET2015. The submission to "your network" is much easier, thanks to the more easy communication and membership spirit. That said, as also answered above, in a first phase it is possible to have a "mix model", but I'm sure that we have 3 groups globally ready for the test.*

I like the idea of a persistent identifier (PID) or something similar. I would envisage that initially the processing would follow the FLUXNET2015 (i.e. ONEFLUX) pipeline and the PID would reflect that. But processing methods do evolve and a Fluxnet steering group could endorse any changes to the pipeline and periodically the PID would change to say FLUXNET2025 for example. It is probably important to think about changes in processing and reprocessing the whole database. This may well happen in the future if we have a new pipeline and you would want to apply the new pipeline retrospectively across all the data I presume? This would be manageable under a single large database but may be difficult under Continental cluster collections.

*The PID identifies the object, the metadata reports the processing. These two together make the data transparent. It is clear that evolution in the processing will happen (hopefully and fortunately) and these will be reflected in the metadata and generate new objects with new PIDs.*

*I agree, in case of new processing a complete reprocessing is needed (like it was between e.g. LaThuile and FLUXNET2015) but it is exactly in cases like this that the Continental clusters are crucial: it is requested an interaction with the PI to get the last version of the data, corrections if needed and then do the processing. A shared workload is the only option with a network growing like FLUXNET. Processing and reprocessing are not only machine time after pushing a button, this could be centralized. There are a lot of interactions needed and this is the key aspect where the Continental clusters are crucial.*

As Papale acknowledges, it will be important full flexibility of each single network to decide what to share and when in FLUXNET and the possibility to distribute different formats and versions through their Data portals. I would envisage that in OzFlux, one could have rapid processing of near real time fluxes using the FLUXNET pipeline on a daily basis and this would have PID that differentiated these data streams as 'beta'

datastreams or something. These data can be used for applications that require real time data but the data comes with the caveats of not being quality controlled by site investigators. The data should be fit for the purpose that it is being applied. As Ray Leuning would tell us, "know thy site" and investigators would still produce the finalised data sets with human skill and site knowledge. So we can have our cake and eat it too. So I think there are potential pros and cons and maybe the best way is to get a working group together and flesh out an operational model for the future. I am very keen and you can count us in.

*Yes, this is exactly the spirit and basis for the system: make each network able to do what they want and at the same time be part (practically, so really feeling this) of FLUXNET. Linking networks to save resources if they want (the continental clusters), gaining visibility and making all the PIs also more engaged in the global networking.*

---

## Author Comment (AC3) · 30 Jul 2020

*Thank you Bruno Marino for the comment to the paper. Here below my replay and view (in blue italic)*

Papale summarizes the history of FLUXNET and related networks resulting in intermittent data products and the current status of mixed interoperability. Papale notes that fluxnet community operations are not standardized, or available in near real-time, or effectively shared creating a mixed-mosaic of data formats and user requirements that are difficult to navigate (see review by JB Fisher). Solutions are proposed to realign flux community/individual efforts and standardization of data format and access, however, few details as to engineering structure and how these solutions could be

[Figure]

implemented are provided. While flux data for CO2 is produced and consumed primarily by research communities, it is of high importance to commercial applications for GHG mitigation and verification of emission reduction claims now emerging in the private sector. Without direct measurement of CO2 flux (e.g., forests, soil, agriculture) at appropriate scales, commercial applications cannot contribute to solutions based on the net removal of GHG's.

*Thanks for the comment. Yes, the private sector is also an important player and although right now the examples of commercial use of FLUXNET data are limited (it is mainly in the sensors production and maintenance), I hope that this will change in the near future, because this would also help FLUXNET to be sustained. The technical and engineering implementation details are not provided, it is true, but I don't think the limit if on that side: there is a lot of experience and development (see for example the ENVRIFAIR activities https://envri.eu/home-envri-fair/) in tools and solutions.*

Adoption and commercialization of typical research instrumentation for eddy covariance will catalyze innovation and cost reduction as well as manufacture of turn-key eddy covariance systems that can be readily deployed anywhere on the planet. Regional networks of up to 1,000 nodes should be readily achievable at reasonable cost. In addition, interoperability should be seamless and draw on a standardized System of Systems architecture design accommodating high data volume, data security, third party verification, and related concept of operations typical of the banking and related industries.

*This is a possibility, and for sure it would be interesting for FLUXNET. The last 25 years shown a development of new sensors and an increase of the networks size, but we need to admit that the technique is still not as accessible as reported in the comment.*

The vision proposed may seem bold and unachievable, however, we live in a time where real-time dynamic data characterizing the biosphere and anthropogenic impacts

on the biosphere are crucial to the management of climate change now and for future generations. Off-the-shelf GHG analyzers, simple steel construction and fabrication of easily deployed single and multiple pole-based towers, and the ubiquitous presence of the internet and satellite telemetry are available and mature industries, poised for application-specific innovation such as proposed by Papale.

*Let's hope this is the near future. . .*

Importantly, private sector involvement may offer new sources of funding for vast eddy covariance observation nodes with the condition that all data remain freely available to the research community.

*This is a crucial point but I think we are still not there. I think it is also our role (FLUXNET) to demonstrate the potential of our measurements also in commercial applications (not only for carbon sequestration but also as service in agriculture, monitoring anthropogenic emissions in cities, linked to satellite validation etc.) to stimulate and investment.*

In summary, Papale proposes that a new "FLUXNET" emerges from a reorganization of existing fluxnet communities. This is a critical transition for FLUXNET, or a likely successor entity, to ensure continued growth and relevance to research and private sectors. The private sector should be actively involved in this transformation benefitting all stakeholders.

*Thank you, I also hope this will be the future direction.*

---

## Author Response (AR1)

Dear Editor and reviewers,
Here as requested a point by point answer (in blue) on the observations and then actions taken to solve the critical points. I think that your comments helped to make the paper more clear, thanks for this.

**COMMENTS FROM EDITOR**

Both Jason Beringer (JB) and Bruno Marion promote important extensions of the Fluxnet data to commercial use and for forecasting which will be good to consider.

These and other suggestions received personally (from people that preferred to not use the discussion channel) have been implemented in the new version of the manuscript

But I am more concerned with the main comments from both JB and Joshua Fisher (JF). While clearly encouraging the discussion that Dario Papale has initiated, there is also a clear dose of skepticism, and a sense of "wishful thinking" (e.g. direct on-line data use, or the expectations with imposing a new world order on clusters). I concur with these concerns.

I was somehow expecting some scepticism, probably also due to a lack of clearness in some part of the paper. I need to underline that the paper has been submitted as "Ideal and perspective", so a personal view based on the past experience in FLUXNET and actual needs that I see. It is clearly not a tentative to impose anything to anybody, I considered this implicit (I can only decide on my personal work). I tried to clarify this in the new version. I think also that probably the fact that in the last 15 years I co-coordinated the FLUXNET synthesis activities introduces a kind of bias in the evaluation of a personal view. This is a problem and I think that exactly what I propose in the paper can help reducing the "personalization" of FLUXNET.

In fact, I was expecting much more community discussion of this 'Discussion Paper' and I, in fact, made a significant effort to get more reviews on board. Failing in both expectations maybe consistent with the concerns raised by the reviewers.

I personally do not agree with the link between number of reviewers and evaluation of the paper. If you look at all the papers in BGD and the amount of comments, the interesting system developed by Copernicus that allows everybody to add comment is (unfortunately) not yet used as we all would like. I also sent emails to FLUXNET and other networks to stimulate discussion and presented the idea in virtual conferences always suggesting to submit their view through the system. However, I received a number of comments personally, from people that did not want to post them publically. I tried to implement also these in the new version.

It seems to me that the comments center on two main arguments. First, the sort of "if it's ain't broke, don't fix it" nature. After all, Fluxnet has been a formidable and enormously successful project (to which author has greatly contributed, and I have been a happy member of for 20 years).

This is true, but also that if there is nothing to fix it does not mean it does not need to/can evolve… The current system worked well because there has been the possibility to invest from single groups for huge efforts in the organization of synthesis activities. The risk is that to see future FLUXNET collections it will be needed to wait a lot of time. In fact, dataset from many people are currently not included in the FLUXNET2015 and there are no plans currently to initiate a new collection. I also think that the proposed system has not so many risks. I tried to revisit the text to make this more clear and analyse them.

The second, seems to suggest "evolution" over the "revolution" of the type proposed, noting some of the dangers, such as losing the data sharing nature of the current system, the complications that can arise from going "continental", and the inability to impose, control, or manage such complicated and totally voluntary system.

The changes proposed would not affect so much the single station that would continue to collect data, submit and get involved, I recognize this was probably not very clear in the paper. It is something affecting more the regional databases that will have to develop more competences and take some of the workload if they are interested to keep FUXNET ongoing. To better clarify this and the fact that it is an evolution and not a revolution I added a new section and a table with the summary of the old vs new tasks for PIs and Regional networks.

The motivation for proposing changes is clear when it comes from someone who invested tremendous efforts in the current system of periodical data compilations. But it might require less committed, done-deal type presentation and more exploration of some alternatives, with more pros and cons.

Here I find some difficulties to find a balance. As said before, this is an Idea and perspective paper, not a formal proposal to other networks or anything forcing anybody. I think I should be free to present my personal view of this based also on the "tremendous efforts" invested and the problems in seeing this as a standard procedure. But see also the two answers below that can contribute to clarify this (I hope)

For example, it could be useful to add a Table comparing the functioning of the existing regional networks, highlighting the good the bad and the missing.

Thanks for the suggestion, I added this and I think it is useful to have a clear picture. It also highlights that the seed for the shuttle is somehow also operational as test case.

It might be good to present some gradual steps that can be taken from the good old system (see both JF and JB major comments) before breaking up to independent clusters. Consider how to gradually expand the already exiting cluster services, but keep the non-aligned sites active partners. Perhaps consider if it would be feasible to maintain the current system and shuttle the responsibility for the periodical data compilation among the sub-networks every few years (an alternative concept to the proposed "Fluxnet Shuttle"?)

Thanks for the suggestion; I added a completely new section on this to propose a possible transition scheme. It does not follow the suggestion of a periodic compilation because, in my opinion, it would not solve the problems and answer the needs (periodic and not continuous, not updated, need to additional resources for the current system, less engagement by the networks) but I tried to suggest a smooth transition path

And so I recommend major revisions of the paper seriously considering the critical (!) comments of the reviewers, going beyond the responses posted so far that are, arguably, not sufficiently convincing. I am confident that a revised version will deserve more response and discussion.

I hope that the revisions are answering the questions and moved the paper more in the suggested direction. All the changes are highlighted in the marked-up manuscript.

**COMMENTS FROM REVIEWER 1 (ADDITIONAL TO THE ONE ALREADY SUBMITTED)**

Isn't this already the *old* organization? I know what you're trying to say here, but it doesn't come across clearly (essentially, those pillars aren't always under the same ceiling, and so it takes some moving of those pillars around in the syntheses, which is time consuming).

I tried to make this more clear adding text and a table, highlighting where are the differences respect to the current system and why these are proposed.

Is it really a new proposal? Hasn't this been proposed for years?

As already said it was not proposed before in these terms, there have been discussions on the need of standardization but not as tool to evolve FLUXNET. I added an example to better clarify where we are now in this process, with the activities in AmeriFlux and ICOS and what is still not discussed and implemented.

Overall, the paper is somewhat light on specific details of how everything would work. I would guess that people are mostly on-board in theory. But, the practical systems engineering could perhaps be flushed out a bit more. Perhaps an additional figure could be useful that would reflect this.

I added a figure and a table to better clarify the data flow. I also added a paragraph where I suggest that all the technical details should be discussed by the networks once/if they agree to implement it (or a different system). This also because as answered above the decisions on implementation and technical and should be defined by the participants (and experts in the specific aspect). I added a reference to activities where these tools are under development.

Are there analog data networks that could be discussed for failures/successes?

I'm not aware of a network with a similar structure (diffuse bottom up initiative without funding) that implemented a similar scheme of coordination; I added a sentence to say that this could be in fact an example for others.

I wonder if FLUXCOM is the best example to justify the proposal. Adding new data to FLUXCOM at this point changes it very little, as far as I would expect, and it moves without real time eddy flux data based on globally gridded inputs. I guess the justification might be better if it were for FLUXCOM-like new initiatives; or, new members to FLUXCOM.

I added a more detailed description on why also for FLUXCOM this is very important and which opportunities could open, being a system that needs continuous recalibration.

I'm trying not to be biased, though I definitely am, but a good example for the remote sensing need was published in Fisher et al. [2020] for the ECOSTRESS mission…     ….We're going to continue to have new missions that would benefit greatly from the proposed global standardized network…

As I already said in the first answer these are very important activities that can benefit from a more accessible and updated FLUXNET and could also strategic for the FLUXNET sustainability. I added references and text on this.

It would be good to include a statement of justification for Continental clusters. (Also, continents are not always consistently defined across the world).

I changed the term and removed the word Continental, changing also the figure and adding a second to better clarify this. I also added a SWOT table to better clarify the benefits.

**COMMENTS FROM REVIEWER 2 (ADDITIONAL TO THE ONE ALREADY SUBMITTED)**

In addition to the demand for real time data that you have outlined, there is also an emerging area of ecological forecasting, for example, The EFI-NEON Research Coordination Network that is an NSF project to create a community of practice that builds capacity for ecological forecasting using NEON. There is also potential demand from data assimilation of flux tower data into short range weather forecast (e.g. https://doi.org/10.1175/MWR-D-19-0370.1). Also real-time agricultural monitoring should be possible. Finally, data can also be used to generate regional real time evapotranspiration estimates using a fusion of flux tower, remote sensing and modelling all of which have potential for use by the public.

Thanks, I added a paragraph to include also these other potential users and applications that are very important and relevant.

With respect to the proposed re-organisation from a single large database into sub collections. On the one hand this helps make the network more sustainable by delegating work and responsibility to continental clusters so that not all the work is being done by a small group. On the other hand this will require continental clusters to be functional, accessible and have open data sharing policies. I would be concerned that many clusters may not have capacity to do this…

I added a section on the transition phase and also clarified how also few clusters could be sufficient to start the implementation of the system. I also removed the word "continental", so that it is clear that this is geographically independent.

…or there may be a difference in data sharing between groups and individuals…

I tried to better clarify with text and a figure that the data policy of individual sites and Regional network doesn't have to change. The point is to agree on the standard once they start to be shared in FLUXNET through the sharing system proposed.

…such that individual sites that may want to contribute are unable because of the inability of the cluster to participate for technical, personnel or other reasons.

I hope this is also more clear now, with the transition phase and the possibility to have a generic entrance for unaffiliated sites under the responsibility of one of the networks (that should provide the funding for this)

Following on from this I can see that there could be many, many sites (even some that were not part of FLUXNET2015) that want to contribute to FLUXNET but they are unable too because they don't have a functional continental cluster. It will be crucial to make sure there is a mechanism for them to contribute.

I added a more clear statement on the importance to get all involved and the proposal for a generic entrance to the system for sites that are not part of any network (although I think that all the sites could and should have a network of reference, because networking with colleagues also regionally is important)

I'm not entirely sure for the rationale and need to move away from a large centralised data base to Continental cluster collections? There has been a lot of effort gone into making the database and I'm not sure if it is broken in some way or has reached its capacity technically. It seems to add another layer of complexity to have Continental cluster collections and a shuttle they queries each of them. It then relies to the ability of the clusters to maintain the data in real time in these clusters. Why not just have a continually updated big database where data is added at any time (daily or annually as it becomes available). You then rely on sites and clusters to push the data continually rather than calling and hassling for sites to submit data (current model). Users can query the data anytime too. We could build a set of tools that allow data to be accessed and queried in more sophisticated ways.

To try to summarize the advantages I added a SWOT table in the new version. I also made more clear that the current system requires resources that don't exist at the moment (organization of a FLUXNET synthesis and the organization of the distribution as it is now cost quite a lot, it should not be charged to only one network or group if we want to raise the level of quality and be robust…).

I like the idea of a persistent identifier (PID) or something similar. I would envisage that initially the processing would follow the FLUXNET2015 (i.e. ONEFLUX) pipeline and the PID would reflect that. But processing methods do evolve and a Fluxnet steering group could endorse any changes to the pipeline and periodically the PID would change to say FLUXNET2025 for example. It is probably important to think about changes in processing and reprocessing the whole database. This may well happen in the future if we have a new pipeline and you would want to apply the new pipeline retrospectively across all the data I presume? This would be manageable under a single large database but may be difficult under Continental cluster collections.

I added explicitly this example (the large massive reprocessing) to explain why the new system would work better and what we can do in case single networks or clusters are not able to do the task.

[revised manuscript text omitted]